# Defining the Effect of Oxytocin Use in Farrowing Sows on Stillbirth Rate: A Systematic Review with a Meta-Analysis

**DOI:** 10.3390/ani12141795

**Published:** 2022-07-13

**Authors:** Sarah V. Hill, Maria del Rocio Amezcua, Eduardo S. Ribeiro, Terri L. O’Sullivan, Robert M. Friendship

**Affiliations:** 1Department of Population Medicine, Ontario Veterinary College, University of Guelph, Guelph, ON N1G 2W1, Canada; shill09@uoguelph.ca (S.V.H.); mamezcua@uoguelph.ca (M.d.R.A.); tosulliv@uoguelph.ca (T.L.O.); 2Department of Animal Biosciences, Ontario Agricultural College, University of Guelph, Guelph, ON N1G 2W1, Canada; eribeiro@uoguelph.ca

**Keywords:** swine, oxytocin, stillbirth, farrowing performance, farrowing duration

## Abstract

**Simple Summary:**

Oxytocin is a hormone that causes smooth muscle contraction and is particularly important during parturition. The administration of exogenous oxytocin to sows at farrowing has been used for decades as a means of assisting sows during parturition. Over the past decade, swine-litter size has dramatically increased, as well as the stillbirth rate. The objective of this systematic review and meta-analysis was to identify the benefits and possible negative side effects of oxytocin use during farrowing. By accumulating data from 25 randomized controlled trials, a meta-analysis examined whether the average number of stillborn piglets, the farrowing duration, and the time interval between the births of piglets were different in the sows that received oxytocin compared to the controls. The results from this study demonstrated that the sows that received oxytocin had an increased average number of stillborn piglets, but experienced reduced farrowing duration and shorter birth intervals between piglets compared to the controls. Future research is required to refine oxytocin usage guidelines, including dosages and the timing of administration. The results of this study demonstrate that it is important to recognize that oxytocin can have adverse side effects, including an increase in stillborn pigs.

**Abstract:**

The objective of this systematic review and meta-analysis was to identify the benefits and possible adverse side effects of oxytocin use during farrowing. Randomized controlled trials that were published in English within the last 50 years were eligible for inclusion. Eligible research needed to contain the PICO elements: population (P)—sows at farrowing; intervention (I):—oxytocin given to sows—comparator (C): sows at farrowing not given oxytocin, as well as sows given different dosages and/or different timing of administration; and outcomes (O):—stillbirths, sow mortality, and piglet viability. Four bibliographic databases were used: PubMed, CAB Direct, Web of Science Core Collection, and ProQuest Dissertations, and Theses Global. In addition, we performed a manual search of the table of contents in the American Association of Swine Veterinarians database for relevant conference proceedings and reports. To assess the risk of bias at the study level, a modified version of the Cochrane 2.0 ROB was used. Meta-analyses were performed to examine the average stillbirth rate, farrowing duration, and birth interval between piglets using random-effect standardized mean difference (SMD) models. To explore heterogeneity, a sub-group analysis was performed on the objectives of the study, dose, time, and route of administration. Of the 46 studies eligible for meta-analyses, only 25 had sufficient information. The pooled analyses of the random effect model demonstrated that the average number of stillborn pigs was lower in the comparator group (SMD = 0.23; CI95% = 0.1, 0.36), and both the farrowing duration (SMD = −8.4; CI95% = −1.1, −0.60) and the birth interval between piglets (SMD = −1.41; CI95% = −1.86, −0.97) were shorter in the oxytocin group. The majority of the studies had an overall risk of bias of ‘some concerns’. It was concluded that the use of oxytocin increases the overall number of stillborn piglets, but decreases the farrowing duration and time interval between piglets. However, future studies should focus on the effect of oxytocin on the experience of dystocia among sows.

## 1. Introduction

According to the 2020 annual PigCHAMP Benchmark report, the average total pigs born per litter was 15.11, with an average number of stillborn pigs per litter of 1.26 [1]. Both of these statistics increased compared to 10 years previously, when the average total pigs born per litter was 13.07 and the average number of stillborn pigs per litter was 0.94 [2]. The relationship between large litter sizes and increased numbers of stillborn pigs is well known. The farrowing of larger litter sizes can have ramifications for both sows and piglets, including extended duration of farrowing, dystocia, intrapartum hypoxia, and an increased risk of a stillbirth [3,4]. The risk of stillbirth is affected by many maternal and fetal factors, including sow age and body condition, as well as piglet weight and birth order. As stillbirth rates rise, it is important to re-evaluate the management practices surrounding farrowing sows.

Oxytocin is an important hormone that is produced in the paraventricular nucleus (PVN) of the hypothalamus and released from the pituitary gland during parturition [5,6]. Oxytocin stimulates myometrial contractions for fetal expulsion and myoepithelial contractions for milk let-down [6]. The administration of exogenous oxytocin has been a common treatment for sows during farrowing for over 50 years [6]. The indications for administering exogenous oxytocin to a sow at farrowing include labor induction when administered in combination with prostaglandin, the acceleration of the normal parturition process, and postpartum uterine debris expulsion [6,7]. The recommended dosage varies widely, with single doses ranging from 10 to 50 units, administered intramuscularly (IM), intravenously (IV), subcutaneously (SQ), or injected into the mucous membrane area of the skinfold of the external vulva.

Over the past decades, oxytocin use in sows has been studied. A survey conducted by the National Animal Health Monitoring System in the U.S. in 1995 found that oxytocin was routinely used on 83.1% of farms [8]. The producers in this survey administered oxytocin for dystocia, poor appetite, mastitis, agalactia, and savaging [8]. In a more recent survey, UK researchers reported that 74% of the respondents used oxytocin at least ‘sometimes’ during farrowing, and 54% of the respondents used oxytocin at least ‘sometimes’ after farrowing [9]. With oxytocin still commonly used, it is important to assess whether it is used in the most beneficial manner with regard to dosage and timing.

The aim of this systematic review and meta-analysis was to acquire and assess all research that has been performed on the administration of oxytocin to peri-farrowing, with the objective of identifying the benefits and possible side effects. The specific questions that were addressed in the systematic review were as follows: (1) Were there negative side-effects if sows received exogenous oxytocin compared to sows that did not receive exogenous oxytocin? (2) When comparing sows that received exogenous oxytocin to sows that did not receive exogenous oxytocin, what was the comparative effectiveness of treatment in reducing stillbirths and improving piglet viability? For questions 1 and 2, the dosage, sow parity, and reason for oxytocin administration were noted.

## 2. Materials and Methods

### 2.1. Eligibility Criteria

Primary studies that were published within the last 50 years (1970–3 July 2020) and in English were eligible for inclusion in this review. Primary studies eligible for inclusion were clinical trials. However, we were interested to know how many observational studies (cross-sectional, cohort, and case-control) there were on the topic. Eligible research needed to contain the PICO elements, as listed below [10,11]:Population (P): Sows either immediately before, during, or immediately after farrowing;Intervention (I): Oxytocin given to sows either immediately before, during, or immediately after farrowing;Comparator (C): Sows either immediately before, during, or immediately after farrowing not given oxytocin—including no treatment, saline, water and/or carbetocin—as well as sows given different dosages and/or different timings of administration (early or late in farrowing process);Outcomes (O): Stillbirths, sow mortality, and piglet viability (such as mortality rate within 2–3 days of birth, meconium staining, ruptured umbilical cords).

### 2.2. Information Sources

Four bibliographic databases were used to search for all eligible primary research, including: PubMed (National Center for Biotechnology Information, Rockville Pike, MD, USA), CAB Direct (Centre for Agriculture and Bioscience International, Wallingford, UK), Web of Science Core Collection (Web of Knowledge, Clarivate, Philadelphia, PA, USA), and ProQuest Dissertations and Theses Global (ProQuest, Ann Arbor, MI, USA) (Appendix A). Following the title/abstract screening process described in Section 2.4, full texts that were not initially acquired from the databases were sought using the University of Guelph’s interlibrary loan (RACER) service. In addition, a manual search in the table of contents for relevant conference proceedings and reports in the American Association of Swine Veterinarians (AASV, Perry, GA, USA) database was performed by a single reviewer. The conferences and reports from the American Association of Swine Veterinarians database included:AASV Annual Meeting (1999–2020);AASV Pre-Conference Seminars (2007–2019);Allen D. Leman Swine Conference (1998–2019);George A. Young Swine Health and Management Conference (1999–2012);International Pig Veterinary Society Congress (2000, 2002, 2004, 2006, 2008, 2010, 2012, 2014, 2016, 2018);International Symposium on Swine Disease Eradication (2001–2002, 2004);ISU Swine Disease Conference for Swine Practitioners (1999–2019);Journal of Swine Health and Production (1993–2020).

The AASV organization office was contacted for further access to conference proceedings and reports if a title and/or abstract was deemed relevant. No corresponding authors were contacted for acquisition of full-text prints or conference proceedings.

### 2.3. Search Strategy

The search strategy comprised 3 distinct components derived from the PICO elements: swine with options focused on mature female pigs (swine, sow$, porcine, pig$, gilt$, sus scrofa, sus domesticus, sus scrofa domesticus), interventions of interest (oxytocin, carbetocin), and important farrowing outcomes (stillbirth, dystocia, farrow*, fetal expulsion, parturition, intrapartum, stillborn, birth, meconium, mortality, hypoxia, fetal death, uterine inertia, parturition complications, perinatal mortality, anoxia, anoxemia, hypoxia, fetal mortality, neonatal mortality) (Appendix A). To ensure all keywords were included, a search for controlled vocabulary terms was performed on MeSH 2020 browser and CAB thesaurus, and all new terms were added to the search string (Appendix A). The search-string terms were connected using Boolean operators ‘AND’ and ‘OR’. All database searches were performed on 23 July 2020 via University of Guelph library resources (Appendix A). All resulting articles were uploaded into a bibliographic software program (EndNote, Clarivate Analytics) and initial duplicate screening was performed. The de-duplicated results were then uploaded onto DistillerSR (Evidence Partners Inc., Ottawa, ON, Canada) for the selection process.

### 2.4. Selection Process

Resulting articles underwent a 2-stage screening process using DistillerSR software. Both the title/abstract and full-text screening forms were pre-tested by the two reviewers. Each screening stage was performed independently by the two reviewers. The first stage assessed the relevance by title and abstract using the following questions:Is the title and abstract available in English?Is the title relevant and is it worth exploring other ways to obtain this paper?Does the title and/or abstract mention exogenous oxytocin or carbetocin use in sows?Does the title and/or abstract mention infectious causes of stillbirths?

The response to these questions included YES, NO, and UNCLEAR. References were excluded if the reader identified a NO response to questions 1–3. References were also excluded if the reader identified a YES response to question 4. If there were any discrepancies between the two reviewers, they discussed their results, and a third reviewer evaluated the title/abstract if the primary reviewers could not reach a consensus. If a reviewer responded with UNCLEAR for any of the questions, both reviewers discussed the situation, and if they could not reach a consensus, a third reviewer evaluated the title/abstract.

The second stage of screening determined the relevance of each article by assessing the full text using the following questions:Is the full text available in English?Is the text more than 500 words?Does the article discuss the use of exogenous oxytocin for farrowing purposes?Does the full text focus on infectious causes of stillbirths?What is the type of study design?Is there a comparator group (either carbetocin, no oxytocin, saline, or water)?

The responses to questions 1–4 and 6 included YES, NO, and UNCLEAR. References were excluded if the reader identified a NO response to questions 1–3 and 6. References were also excluded if the reader identified a YES response to question 4. The responses to question 5 included ‘review article or systematic review’, ‘commentary without original results’, ‘the intervention is not the main focus of the study (e.g., only mentioned in the discussion or references)’, ‘observational study’, ‘clinical trial’, ‘unsure’, and ‘case report’. References were excluded if the reviewer identified the reference as ‘review article or systematic review’, ‘commentary without original results’, ‘the intervention is not the main focus of the study’, or “case report”. Clinical trials and observational studies were included and further categorized by study type. If there were any discrepancies between the two reviewers, the matter was discussed, and if the reviewers could not reach a consensus, a third reviewer evaluated the full text.

### 2.5. Data Collection Process

The eligible studies were further processed for data collection. Two reviewers independently filled out a standardized form using DistillerSR (Evidence Partners Inc., Ottawa, ON, Canada). If there were discrepancies in answers between the two reviewers, a discussion took place in order to reach an agreement, and if the reviewers could not reach a consensus, a third reviewer decided.

### 2.6. Data Items

#### 2.6.1. Study and Population Characteristics

Study characteristics extracted included year of publication, year (or range) in which study was conducted, month(s) during which study was conducted, country in which study was conducted, study design, objective of the study, number of study groups, type of allocation to groups, number of replicates in study per group, number of sites, and whether blinding was used for the administration of treatment. Population characteristics extracted included type of herd, breed of sows, number of sows included in the study, and parity of sows.

#### 2.6.2. Intervention and Comparator Characteristics

Comparators included carbetocin and any type of placebo, such as water and saline. Carbetocin was grouped as a comparator due to pharmacological differences to oxytocin. While carbetocin is a synthetic analogue of oxytocin, it has a half-life roughly 10 times greater than oxytocin (~40 min versus ~4 min) [12,13,14,15]. Intervention and comparator characteristics extracted included description of the group, the product used (i.e., brand), dosage, number of doses and the interval between doses, route and site of administration, dosage of prostaglandin, other treatments used, number of sows from each parity (1, 2–5, 6+), reason for oxytocin use, whether producers performed manual assistance and when, and number of sows that needed manual assistance.

#### 2.6.3. Eligible Outcome

Outcome characteristics for both intervention and comparator groups extracted included:Average duration of farrowing (in minutes/hours);Average birth interval between piglets (in minutes);Average total piglets born, average number of piglets born alive, average number of piglets stillborn;Total number of piglets born, number of piglets born alive, and number of piglets stillborn (raw and/or proportion);Total number of sows that had at least one stillborn pig (raw and/or proportion);Farrowing order of stillborn pigs;Average time sows spent standing (in minutes), lying down (in minutes), eating/drinking (in minutes), nesting (in minutes);Number of piglets with meconium staining;Number of piglets with mild, moderate, and severe meconium staining;Number of piglets with ruptured umbilical cords at birth;Number of sow mortalities (raw and/or proportion).

### 2.7. Risk of Bias in Individual Studies

The Risk of Bias 2 tool (RoB 2.0), a modified version of the Cochrane tool for risk of bias for randomized studies of interventions, was used for assessing the risk of bias for each eligible randomized trial [16,17,18]. The assessment was performed at the study level; therefore, a RoB analysis was performed on each trial presented in the articles. The RoB assessment tool contains a set of 5 domains used to assess the overall risk of bias: bias arising from the randomization process, bias due to deviations from intended interventions, bias due to missing outcome data, bias in measurement of the outcome, and bias in the selection of the reported results [18]. Each domain consisted of a series of signaling questions, and, based on the response, the overall risk of bias was determined as ‘Low’, High’, or ‘Some concern’ [18]. The overall risk of bias was considered ‘Low’ if all domains were ‘Low’; ‘High risk’ was determined if any domain had at least 1 ‘High risk’; and ‘Some concern’ was determined if any domain had at least 1 ‘Some concern’ and no ‘High risk’ [18].

Two reviewers independently filled out a standardized risk-of-bias form on Microsoft Excel (Version 16.54, Microsoft, Redmond, USA) for each study. If there were discrepancies in answers between the two reviewers, discussion was held to form an agreement, and if the reviewers could not reach a consensus, a third reviewer decided. Visualization of the resulting risk of bias was performed using R version 3.6.1, ‘Action of Toes’, in Rstudio version 1.3.1093, ‘Apricot Nasturtium’ (R Foundation for Statistical Computing, Vienna, Austria), with the package robvis [19,20].

### 2.8. Effect Measures

For the synthesis of results, the effect measures used were mean and standard deviation. Results were presented as meta-analysis with a standard mean difference (SMD) and 95% confidence interval (CI95%).

### 2.9. Synthesis Methods

The arm level data collected for each eligible outcome were sum, mean with standard deviation, standard error or 95% confidence interval, and median in any units. In an eligible study, groups that were administered oxytocin were categorized as the intervention group (e). Study groups that received saline, water, nothing, or carbetocin were categorized as the comparator group (c). Further, if an eligible study had multiple trials and/or more than 2 study groups, and the sub-group fell into either category, the data were collected. Therefore, some eligible studies had multiple contributions to the meta-analyses.

Results were categorized into 3 outcomes: stillbirth, farrowing duration, and birth interval between piglets. Descriptive statistics using Stata/IC 16.1 for Mac (StataCorp, College Station, TX, USA) included: total number of studies; number of independent outcomes in the intervention groups subdivided into the total number of sows that received: 1. oxytocin or carbetocin, 2. dose and route used for each intervention; number of independent outcomes in the comparator group; and total number of sows in each comparator group.

For the meta-analysis, each study group needed a mean and standard deviation. Studies that presented stillbirth data as averages and standard errors on a percentage scale were transformed (Equations (A1)–(A3) in Appendix B). Standard errors were transformed into standard deviations (Equation (A4)) using methods described by the Cochrane Handbook for Systematic Reviews of Intervention [21]. Studies that presented CI95% were transformed into a standard deviation (Equations (A5)–(A7)) using methods described by the Cochrane Handbook for Systematic Reviews of Intervention [21]. In addition, all means and standard deviations needed to be in the same unit. For example, farrowing duration data presented in hours were transformed into minutes (Equations (A8) and (A9)). 

To perform a quantitative summary of results for stillbirth, farrowing duration, and birth interval between piglets, a meta-analysis was performed using R version 3.6.1, ‘Action of Toes’, in Rstudio (Version 1.3.1093, R Foundation for Statistical Computing, Vienna, Austria) [19]. The packages used for uploading and downloading data were readxl and xslx, respectively [22,23]. For data clean-up and formatting, the package tidyverse was used [24]. Meta package was used to perform the meta-analyses and to create forest plots [25]. The meta-analytical method included a random-effects model, which incorporates the assumption that different studies measure the results of interest slightly differently. The random-effects model incorporated DerSimonian-larid estimator for tau2, which was used to assess the between-study variance [25]. The Jackson method was used for confidence interval of tau2 and tau [25]. Lastly, Hedges’ g was used to measure the bias-corrected standardized mean difference [25].

To explore heterogeneity between studies, sub-group analysis was explored for each of the outcomes of interest. A sub-group analysis was performed on study objectives ‘induction program’ and ‘farrowing assistance’. A separate meta-analysis was performed on each study objective, further analyzing the dosage, time of administration, and route of administration.

### 2.10. Reporting Bias Assessment

The package ‘metafor’, using R version 3.6.1, ‘Action of Toes’, in Rstudio (Version 1.3.1093, R Foundation for Statistical Computing, Vienna, Austria)’, was used to assess bias [19,26]. Bias assessment was performed on each of the outcomes assessed: mean stillbirth, farrowing duration, and time interval between piglets. To assess symmetry, a funnel plot was presented as illustration and the Egger’s line-of-regression test was conducted to perform a linear regression test of funnel-plot asymmetry. A null hypothesis (*p* > 0.01) was that there is symmetry and no publication bias. The alternate hypothesis (*p* < 0.01) was that there is asymmetry and publication bias. If there was asymmetry, a trim-and-fill approach was conducted to adjust the meta-analysis for publication bias.

### 2.11. Certainty Assessment

The strength of evidence for each outcome was assessed using the Grading of Recommendations Assessment, Development and Evaluation (GRADE) methodology [27,28]. This system assessed the risk of bias, publication bias, precision, directness, and consistency for each outcome [27,28].

## 3. Results

### 3.1. Study Selection

A total of 618 unique references were acquired from all the databases; 285 remained after the title/abstract screening, and 58 studies were deemed eligible after the full-test screening (Figure 1). Three hundred and thirty-three studies were excluded during the title and abstract stage for the following reasons: they were not written in English (9); there was no mention of the administration of exogenous oxytocin or carbetocin to sows (310); the article mentioned infectious causes of stillbirths (2); no abstract or full text were available (12). During the full-text stage, studies were excluded for the following reasons: the full text was not available in English (161); the text was <500 words (9); the text did not discuss the use of exogenous oxytocin for farrowing purposes (30); the focus of the text was infectious causes of stillbirths (1); the study design was a review article or systematic review (11); the study was a commentary without original results (5); the intervention was not the main focus (2); the paper was a case report (1); or there was no comparator group (6). One paper was excluded during the screening process due to concerns over its questionable experimental methods. Among the 58 studies deemed eligible for data extraction, the cross-sectional (11) and cohort (1) studies were not further processed. Therefore, 46 clinical-trial studies were eligible for data collection.

### 3.2. Study Characteristics

Less than half of the studies reported the country in which the trial was conducted (*n* = 18/46; 40%). These studies were conducted in eight different countries, with the most commonly reported country being Mexico (*n* = 7/46; 15%), followed by Canada (*n* = 3/46; 6%) (Appendix A). The year when the study was conducted was not reported in most of the studies (*n* = 34/46; 74%). The studies that reported when the trial was conducted were between 1984 and 2005. The majority of these studies were conducted on commercial farms (N = 24/46; 52%) and research farms (*n* = 7/46; 15%) (Appendix A). However, a third of the studies did not specify the type of facility where the research was conducted (*n* = 16/46; 35%). The studies were categorized into two groups based on the research objectives: induction program (*n* = 28/46; 60%) or farrowing assistance and dystocia (*n* = 18/46; 39%) (Appendix A). Among the 46 papers, some reported the results of multiple trials, including two trials (N = 5/46; 11%), three trials (*n* = 1/46; 1.8%), and four trials (*n* = 1/46; 1.8%). Only 25 papers (*n* = 25/46; 54%) had sufficient information to be used in a meta-analysis. Based on the data extracted, the most common outcomes reported were stillborn pigs per litter (*n* = 19/46; 43%), farrowing duration (*n* = 22/46; 50%), and birth interval between piglets (*n* = 11/46; 24%) (Table 1). The majority of the studies had more than two study groups that were relevant to either the intervention or the comparator, and the outcomes from every relevant study group were included.

### 3.3. Risk of Bias in Studies

All the studies were assessed using the modified version of the Cochrane tool for the risk of bias for randomized studies of interventions (RoB 2.0) [17,18]. An assessment was performed on individual trials for each trial. Among the 46 references, risk-of-bias assessments were performed on a total of 56 trials. The most common outcome for bias arising from the randomization-process domain was ‘some concern’ (*n* = 51/56; 91%) Appendix A. Two trials were deemed ‘low’, and 3 were deemed ‘high’. The most common outcome for bias due to deviations from the intended-interventions domain was “low’ (*n* = 48/56; 86%), followed by ‘some concern’ (*n* = 7/56; 12.5%), and ‘high” (*n* = 1/56; 1.8%). For the domain of bias due to missing outcome data, the outcome was split between ‘low’ (*n* = 34/56; 61%) and ‘some concern’ (*n* = 22/56; 39%). For the domain of bias in the measurement of outcomes, the majority of the trials were ‘some concern’ (*n* = 53/56; 95%), and the remaining few were ‘low’ (*n* = 3/56; 5.3%). The most common outcome for the bias in the selection of the reported results domain was ‘low’ (*n* = 37/56; 66%), and the rest of the trials were ‘some concern’ (*n* = 19/56; 34%). The overall risk of bias for the majority of the trials was ‘some concern’ (*n* = 51/56; 91%) followed by ‘high’ (*n* = 4/56; 7%). One trial had an overall “low’ risk of bias.

### 3.4. Results of Individual Studies

Twenty-four references had sufficient information for a meta-analysis on the average number of stillborn pigs per litter, farrowing duration, and/or birth interval between piglets. The results of the individual studies were further examined by categorizing them into the three outcome variables.

#### 3.4.1. Stillbirth

There were 50 independent group means and SD for stillbirth in the intervention study group (50 comparisons from 19 studies). Most of the studies investigated the effects of oxytocin (46/50; 92%), and a small number studied carbetocin (4/50; 8%). The most common method of treatment administration was intramuscular (IM) (46/50; 92%), followed by intravenous (2/50; 4%) and intravulvar (IVU) (2/50; 4%). The most common dosage used for oxytocin was “based on weight” (11/46; 24%). The lowest dosage used was 5 IU (3/46; 6.5%), followed by 10 IU (10/46; 22%), 20 IU (5/46; 11%), 25 IU (4/46; 8.6%), 30 IU (9/46; 20%), and 40 IU (4/46; 9%). The comparator study group included 36 independent means and SD for the stillbirth outcomes (36/19 studies), including no treatment (17/36; 47%), saline (16/35; 44%), and water (3/36; 8%). The mean number of stillborn pigs per litter in the intervention group was 0.81 ± 0.42 (*n* = 1790 sows) and the mean number of stillborn pigs per litter of the comparator group was 0.69 ± 0.36 (*n* = 1321 sows).

#### 3.4.2. Farrowing Duration

There were 51 independent group means and SD for farrowing duration in the intervention study group (51/22 studies), most of the studies investigated the effects of oxytocin (45/51; 88%), and a small number studied carbetocin (6/51; 12%). The most common method of treatment administration was intramuscular (IM) (47/51; 92%), followed by intravenous (2/51; 4%) and intravulvar (IVU) (2/51; 4%). The most common dosage used for oxytocin was “based on weight’ (14/45; 31%). The lowest dosage was 5 IU (3/45; 6.6%), followed by 10 IU (8/45; 17.8%), 20 IU (7/45; 15.5%), 25 IU (4/45; 8.9%), 30 IU (4/45; 8.9%), and 40 IU (5/45; 11%). The comparator study group included 37 independent group means and SD for farrowing duration (37/22 studies), including: no treatment (20/37; 54%), saline (16/37; 43%), and water (1/37; 2.7%). The mean farrowing duration in minutes in the comparator group was 229 ± 55 (*n* = 2182 sows), and the mean farrowing duration of the intervention group was 183 ± 51 (*n* = 2532 sows).

#### 3.4.3. Birth Interval between Piglets

There were 23 independent group means and SD for the birth interval between piglets in the intervention study group (23/11 studies). Most of the studies investigated the effects of oxytocin (22/23; 96%), and a small number studied carbetocin (1/23; 4.3%). The most common method of treatment administration was intramuscular (IM) (19/23; 83%), followed by intravenous (2/23; 8.7%) and intravulvar (IVU) (2/23; 8.7%). The most common dosage used for oxytocin was ‘based on weight’ (14/22; 64%). The lowest dosage was 10 IU (3/22; 13.6%), followed by 20 IU (2/22; 9.1%) and 40 IU (3/22; 3%). The comparator study group included 14 independent group means and SD for the time interval between piglets 14/11 studies), including no treatment (3/23; 13%), saline (10/23; 43.5%), and water (1/23; 4.3%). The mean interval between piglets in minutes for the comparator group was 22.1 ± 4.5 (*n* = 643 sows), and the mean time interval between piglets for the intervention group was 15.5 ± 5 (*n* = 1096 sows).

### 3.5. Results of Syntheses

#### 3.5.1. Stillbirth

An initial meta-analysis examined the mean difference between the stillbirths from 19 studies, including studies with more than two study groups. A total of 46 oxytocin intervention groups were used in comparison with 36 control groups (Figure 2a). The pooled analysis using the random-effects model demonstrated that the control groups had, on average, 0.23 fewer stillborn pigs per litter compared to the oxytocin group (SMD = 0.23, CI95% = 0.10, 0.36). There was a variability of 70.4% across the studies due to between-study variation (I^2^ = 70.4%, CI95% = 60.3%, 78.0%, *p* < 0.01). Further investigation was performed to examine the heterogeneity using a sub-group analysis by objective categorization: induction program and farrowing assistance (Figure 2b). For the induction-program sub-group, there were 29 study groups, and a pooled analysis demonstrated that the increase in stillborn pigs in the oxytocin treatment group (0.085 more stillborns) was not substantially different from that in the control group (SMD = 0.085, CI95% = −0.04, 0.21). The variability across the studies due to the low between-study variation (I^2^ = 30%, *p* = 0.07). The farrowing-assistance sub-group consisted of 18 study groups, and the pooled analysis demonstrated the control group had an average number of stillborn pigs per litter that was smaller than the oxytocin-treated group (SMD = 0.42, CI95% = 0.20, 0.64). The heterogeneity between the sub-groups was significant (Q = 6.48, *p* = 0.01). Further analyses on the dose, route, and time of administration were all performed according to the study objectives.

The sub-group analyses examining the implications of dosage on the average stillbirths for the induction-program studies was variable and insignificant (Figure 3a). The majority of the pooled analyses by timing of administration were also insignificant (Figure 3b). However, among the 10 studies that administered oxytocin 20 h after prostaglandin (or analogue) as the treatment group, there were fewer stillborn pigs per litter in the control group (SMD = 0.18, CI95% = 0.003, 0.35, I^2^ = 17%). A sub-group analysis of the route of administration was not performed for the induction-program studies, since almost all these studies used an intramuscular injection for oxytocin administration (*n* = 28/29; 97%).

A sub-group analysis examining the implications of dosage for the average number of stillborn pigs per litter in farrowing-assistance studies found mostly insignificant differences between treated subjects and controls (Figure 4a). However, the studies in which oxytocin was administered using a dosage that was dependent on individual sow weight demonstrated that the control group was more likely to have a reduced number of stillborn pigs per litter (*n* = 11, SMD = 0.38, CI95% = 0.15, 0.61, I2 = 73%). The majority of the pooled analyses for the timing of administration were significant (Figure 4b). The administration of treatment after the expulsion of the first piglet had a moderate effect (*n* = 16, SMD = 0.35, CI95% = 0.25, 0.45, I^2^ = 80.9%). The control group was more likely to have a reduced number of stillbirths in each sub-group of the route of administration, IM (*n* = 43; SMD = 0.32, CI95% = 0.08, 0.55, I^2^ = 83.7%), and IV (*n* = 2; SMD = 0.92, CI95% = 0.61, 1.22, I^2^ = 10.8%) (Figure 4c).

#### 3.5.2. Farrowing Duration

An initial meta-analysis examined the mean difference in farrowing duration from 21 studies, including studies with more than two study groups, and a total of 46 oxytocin intervention groups versus control groups were used (Figure 5a). The pooled analysis using the random-effects model demonstrated that the intervention group had a reduced farrowing duration compared to the control group (SMD = −8.4, CI95% = −1.07, −0.60). There was a variability of 92.6% across the studies due to between-study variation (I^2^ = 92.6%, CI95% = 91.0%, −94.0%, *p* < 0.01). One study was assessed as having a ‘high risk’ of bias. A form of sensitivity analysis was performed: a meta-analysis was performed without the ‘high-risk’ study and a comparison was drawn. The pooled analysis using the random-effects model demonstrated a similar finding to the analysis that excluded the study (SMD = −7.8, CI95% = −0.98, −0.59). There was a variability of 91.9% across the studies due to between-study variation (I^2^ = 91.9%, CI95% = 90.2%, 93.04%, *p* < 0.01). All further analyses were performed with all the articles.

Further investigation was performed to examine the heterogeneity using a sub-group analysis by objective categorization: induction program and farrowing assistance (Figure 5b). For the induction-program sub-group, there were 21 study groups, and a pooled analysis demonstrated that the intervention group reduced the farrowing duration compared to the control group (SMD = −0.27, CI95% = −0.46, −0.08). There was variability across the studies due to between-study variation (I^2^ = 68%, *p* < 0.01). The farrowing-assistance sub-group consisted of 25 study groups, and the pooled analysis also demonstrated that the intervention group was strongest, providing reduced the farrowing duration (SMD = −1.29, CI95% = −1.64, −0.92; I^2^ = 95%, *p* < 0.01). The heterogeneity between the sub-groups was significant (Q = 23.64, *p* < 0.001).

#### 3.5.3. Birth Interval between Piglets

An initial meta-analysis examined the mean difference in birth intervals between piglets across 11 studies (including studies with multiple study groups), with a total of 22 oxytocin-intervention and control groups (Figure 6). The pooled analysis using a random-effects model demonstrated that the intervention group had a reduced birth interval between piglets compared to the control group (SMD = −1.41, CI95% = −1.86, −0.97). There was a variability of 94.9% across the studies due to between-study variation (I^2^ = 94.9%, CI95% = 93.4–96.1%, *p* < 0.01). Due to insufficient numbers, a sub-group analysis on objectives was not performed for this outcome. Only two results examined the induction programs, and both came from a single reference [29].

### 3.6. Reporting Biases

A bias assessment was performed on each of the outcomes assessed: mean stillbirth, farrowing duration, and time interval between piglets.

#### 3.6.1. Stillbirth

Based on the visual assessment of the contour-enhanced funnel plot and Egger’s line-regression test, we did not reject the null hypothesis (*p* = 0.06) and concluded symmetry (Appendix A). Therefore, there was no significant publication bias detected in the analysis of the mean number of stillborn pigs per litter.

#### 3.6.2. Farrowing Duration

Based on the visual assessment of the contour-enhanced funnel plot and Egger’s line-regression test, we did not reject the null hypothesis (*p* = 0.08) and concluded symmetry (Appendix A). Therefore, there was no significant publication bias detected in the analysis of the farrowing duration.

#### 3.6.3. Interval between Piglets

Based on the visual assessment of the contour-enhanced funnel plot and Egger’s line-regression test, we did not reject the null hypothesis (*p* = 0.038) and concluded symmetry (Appendix A). Therefore, there was no significant publication bias detected in the analysis of the time interval between piglets.

### 3.7. Certainty of Evidence

Based on the strength of the evidence using GRADE guidelines, our results indicated moderate quality [27,28]. Each outcome was downgraded by one in the indirectness domain due to ‘Indirect comparison to dose’ (Appendix A). Based on the definition of the moderate rating for the certainty of evidence, ‘This research provides a good indication of the likely effect. The likelihood that the effect will be substantially different is moderate’ [28].

## 4. Discussion

The administration of exogenous oxytocin to sows is a common farming practice for inducing farrowing following prostaglandin, for accelerating the normal parturition process, and for the expulsion of placental debris postpartum [6,7]. Induction programs consist of the administration of prostaglandin (or analogue) on day 111–114 of gestation and, in some cases. Oxytocin is also administered around 20–24 h after the prostaglandin injection [54]. Prior to the expulsion of piglets, the preparation of labor involves a decrease in progesterone and a spike in prostaglandin [54,55,56]. This leads to a cascading effect, resulting in an increase in relaxin and estrogen, which results in cervical dilation [54,55,56]. During this time, oxytocin receptors are expressed and myometrial contractions commence [54,55,56].These programs may be considered advantageous in large herds because the timing of farrowing can improve labor and farm-management efficiency [54]. The synchronization of farrowing increases the proportion of births during working hours, creating a more efficient use of time and labor for herdsmen [54]. Even if sows are not induced to farrow with prostaglandins, oxytocin may be used to shorten the duration of farrowing to facilitate piglet-management practices, such as cross-fostering.

A considerable amount of research has examined the use of oxytocin in sows to determine its benefits and implications for the farrowing process and stillbirth rates [5]. Endogenous oxytocin stimulates myometrial contractions, which help piglets exit the birth canal, resulting in a positive feedback loop, known as the Ferguson reflex [55,57]. Administering oxytocin increases the amplitude and frequency of myometrial contractions [55,57]. In this review, it was noted that the dosage, route of administration, and timing of the treatment varied considerably from study to study, reflecting the variation in oxytocin use on farms [55,57]. One aim of this review was to create more specific guidelines for oxytocin use for producers by performing meta-analyses to assess its effects on stillbirths, farrowing durations, and birth intervals between piglets.

The studies examining the use of oxytocin for farrowing assistance demonstrated that overall oxytocin was not effective in reducing the number of stillborn pigs compared to controls. Oxytocin reduced the mean farrowing duration and birth interval. Although these findings were statistically significant, there was some variability across the studies due to between-study variation. After examining the meta-analyses on the effects of dosage, timing of injection with respect to onset of farrowing, and route of administration on the mean number of stillborn pigs, consistently, the control group had fewer stillborn pigs compared to sows that received oxytocin. However, it is important to note that there was heterogeneity between the studies. This highlights the lack of consistency in the results due to the large variation in the practical usage of oxytocin. Varying dosages, timing, and routes of administration result in a larger set of outcomes. In this review, 76% of the studies administered a standard dosage per sow, which varied from 5 IU to 40 IU rather than use a dosage based on the weight of the sow. This probably reflects the common practice in the field, but it should be noted that, typically, in a group of farrowing sows, body weight may vary from sow to sow by more than 100 kg. Likewise, a systematic review examining uterotonics in sows presented similar conclusions with regard to dosage variation [58]. Further research should be undertaken to examine whether stillbirth associated with the use of oxytocin can be reduced by matching the dosage with the weight of the sow in order to prevent under- or over-dosing [58].

Of all the studies in this review, 60% examined the incorporation of oxytocin in induction programs, and a large proportion studied the administration of oxytocin within 20–24 h of the prostaglandin (or analogue). In the meta-analysis examining stillbirths, the sub-group of studies that received oxytocin 20 h after the prostaglandin (or analogue) had an increased average stillbirth rate compared to the controls. It has been noted in previous research that there is a concern regarding oxytocin administration at a standard time, since the cervix might not be fully dilated, resulting in an increased risk of dystocia, further jeopardizing the viability of piglets [32,54,55]. When oxytocin is administered early in the parturition process, particularly at high dosages, there is an increased number of pigs born with ruptured umbilical cords, and there is a danger of strong uterine contractions interfering with the blood flow to the placenta [32,54,55].

Ninety-one percent of the risk-of-bias assessments at the study level were of ‘some concern’. These findings are consistent with the fact that reporting guidelines are relatively new in research on livestock and food safety. The Reporting Guidelines for Randomized Controlled Trials for Livestock and Food Safety (REFLECT) was not published until 2010 [59,60]. REFLECT is a reporting checklist that provides guidelines to ensure improved study reporting [59]. During the risk-of-bias assessment for this review, it was noted that there was a lack of transparency with regard to the exact methodology [59]. For example, although studies would state that they assigned sows by ‘random allocation’, they did not elaborate on how it was random (i.e., table, random number generator). This was also noted in previous research assessing the evidence of improved reporting in swine-vaccination trials post-REFLECT-statement [59]. None of the studies that were examined included all the REFLECT items, and it was concluded that there was room for improvement in reporting [59]. It is interesting to note that, during the article-screening process in this review, of the 618 unique references, only 12 observational studies were deemed relevant to our PICO research questions. However, further investigations of observational studies were not performed, as the meta-analysis only consisted of clinical trials. Clinical trials are deemed the gold standard for the investigation of interventions as they reduce bias and minimize potential confounders [61]. However, the main disadvantage of clinical trials is their lack of external validity and generalizable results [61,62].

In this meta-analysis, the assumption of the population of sows was that they were eutocic. However, two articles in this review that were not included in the meta-analysis examined the use of oxytocin on sows that experienced dystocia [63,64]. The authors defined a dystocia event as a sow that had at least one stillborn piglet among its first four births [63,64]. They found that the dystocic sows that received oxytocin after the^ir^ fifth piglet had approximately 50% fewer stillborns compared to the dystocic sows that did not receive oxytocin [63,64]. The researchers also found that in both the eutocic and dystocic sows that received oxytocin, approximately 50% fewer piglets were born with anoxia compared to piglets born to sows not administered oxytocin [63,64]. A more recent cross-sectional study, which defined dystocia as a birth interval exceeding 45 min or the use of obstetric intervention, found that 47.2% of the 387 sows experienced dystocia [65]. The same authors reported that among the 86.5% of sows that received one dose of oxytocin, the median birth for which it was administered was the eighth piglet [66]. It was also reported that 60% of the sows in the study had at least one stillbirth. It was suggested that there is an association between increased birth order and stillbirths rather than a relationship between oxytocin use and the increased likelihood of a stillbirth [66]. The practical implication of this review is that the administration of to sows early in the farrowing process regardless of they experience difficulties results in shorter farrowing, but increases the likelihood of a stillborn piglet, whereas oxytocin given to sows with dystocia later in the farrowing process probably results in fewer stillborn piglets compared to no intervention. However, further investigation into the use of oxytocin focused on treating sows with dystocia needs to be performed to provide a clearer understanding of the benefits and risks of using oxytocin in the farrowing room. Given the importance of assessing interventions in their natural settings, it is suggested that more observational studies should be performed [67,68].

## 5. Conclusions

This review highlighted that while oxytocin has been used in swine production for decades, and has been studied extensively, there are gaps in the literature that are worth exploring. The meta-analyses demonstrated that the sows that received oxytocin had an increased number of stillborn piglets per litter, but reduced farrowing durations and birth intervals between piglets compared to the control sows. Further research is required in the future to refine oxytocin-usage guidelines, including dosages and the timing of administration. The results of this study demonstrate that it is important to recognize that oxytocin can have adverse side effects.

## 6. Registration and Protocol

A protocol was established a priori, following the Preferred Reporting Items for Systematic review and Metal Analysis Protocols (PRISMA-P) reporting guidelines. The protocol was registered with the University of Guelph’s institutional repository (https://atrium.lib.uoguelph.ca/xmlui/handle/10214/18115) (accessed on 28 July 2020).

*Amendment 1:* Include the following to the reference list: [6,9,69].

*Deviation 1:* For the title/abstract screening form, a question was added (‘Is the title relevant- is it worth exploring other ways to get this paper?’). This was to include references whose title seemed relevant, but the abstract was unavailable. If the response to this question was ‘NO’, then the reference was excluded. If the response to this question was ‘YES’ or ‘UNCLEAR’, then the references were included and University of Guelph RACER was used to acquire the full-text article.

*Deviation 2:* For the full-text screening form, the question ‘Is the study design a trial?’ was changed to ‘Type of study design?’. This is because we wanted to determine how many observational studies were relevant to our research question.

*Deviation 3:* For the full-text screening form, the question ‘Is there a comparator group (either carbetocin or no oxytocin)?’, the wording was changed to ‘Is there a comparator group (either carbetocin, another oxytocin analogue or no oxytocin)?’. This was to ensure that the question was more transparent.

*Deviation 4:* The risk-of-bias assessment was performed at the ‘study’ level instead of the ‘outcome’ level. The change was due to the manner in which the references reported their methodology and results.

*Deviation 5:* The meta-analyses did not include carbetocin due to the possibility of variation with other control substances (saline, water, and nothing).

*Post-protocol decision:* The sensitivity analysis was performed for the outcome of the farrowing duration because a single study had a different risk-of-bias assessment result.

*Post-protocol decision:* Sub-group analyses by objective, dose, time, and route of administration were performed to investigate heterogeneity.

References included in qualitative analysis are presented in Appendix A [70,71,72,73,74,75,76,77,78,79,80,81,82,83,84,85,86,87,88,89,90,91,92].

## Figures and Tables

**Figure 1 animals-12-01795-f001:**
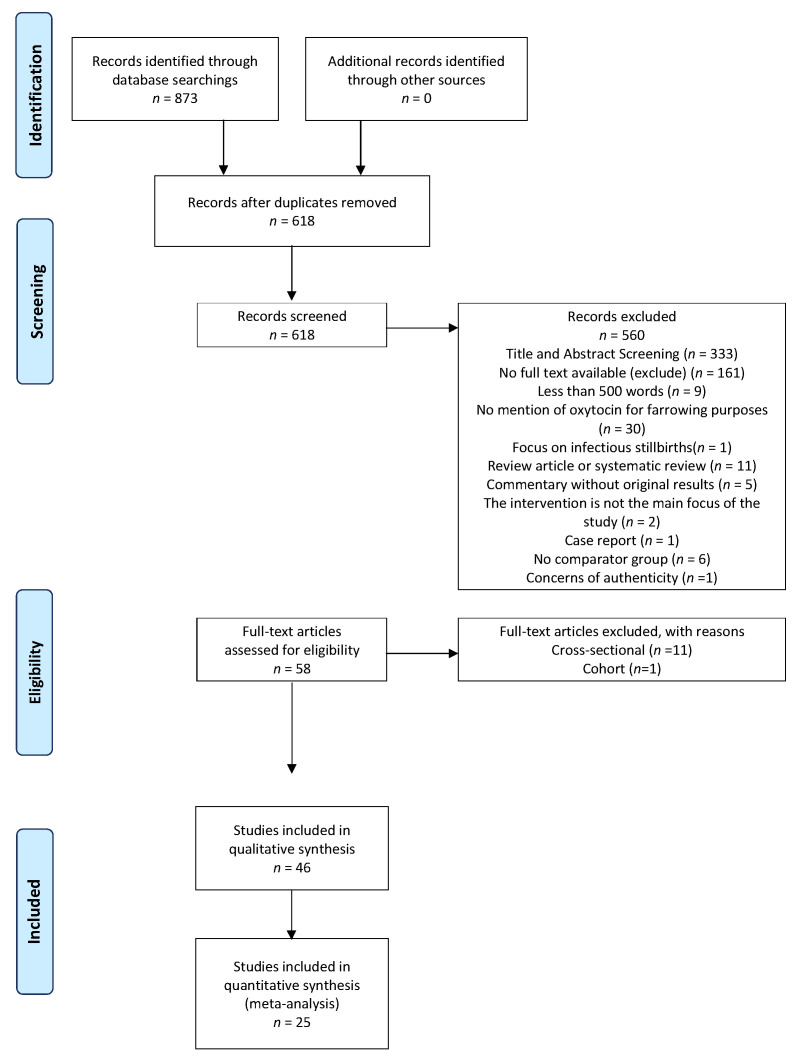
Preferred reporting items for systematic review and meta-Analysis (PRISMA) flow diagram.

**Figure 2 animals-12-01795-f002:**
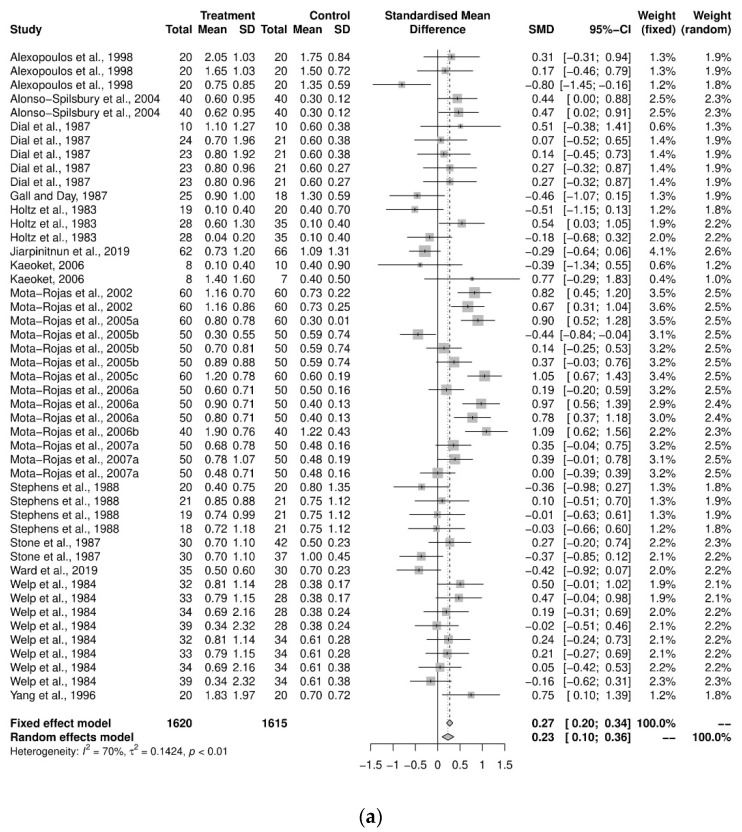
(**a**) Meta-analysis examining mean stillbirths: pooled analysis [29,30,33,35,36,37,38,39,40,41,42,43,44,45,48,49,51,52,53] (**b**) Sub-group analysis by study objectives (induction program and farrowing assistance) [29,30,33,35,36,37,38,39,40,41,42,43,44,45,48,49,52,53].

**Figure 3 animals-12-01795-f003:**
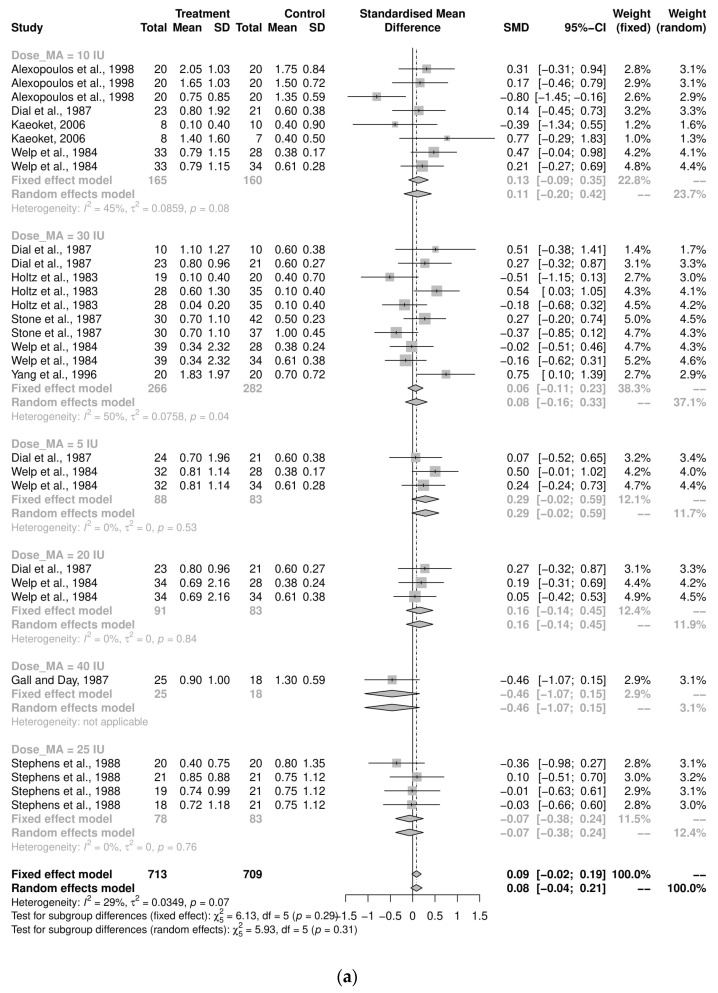
(**a**) Meta-analysis examining mean stillbirths in induction-program studies: Sub-group analyses by dosage [29,33,35,36,38,48,49,52,53]. (**b**) Meta-analysis examining mean stillbirths in induc-tion-program studies: Sub-group analysis by timing of administration [29,33,35,36,38,48,49,52,53].

**Figure 4 animals-12-01795-f004:**
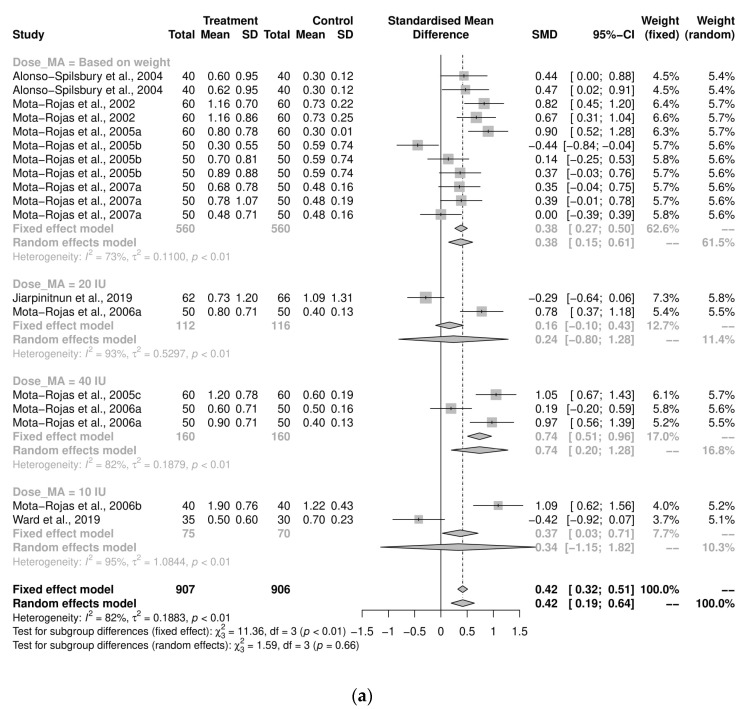
(**a**) Meta-analysis examining mean stillbirths in farrowing-assistance studies: sub-group analyses by dosage [30,37,39,40,41,42,43,44,45,51]. (**b**) Meta-analysis examining mean stillbirths in farrowing-assistance studies: sub-group analysis by timing of administration [30,37,39,40,41,42,43,44,45,51]. (**c**) Meta-analysis exam-ining mean stillbirths in farrowing assistance studies: sub-group analysis by route of administration [30,37,39,40,41,42,43,44,45,51].

**Figure 5 animals-12-01795-f005:**
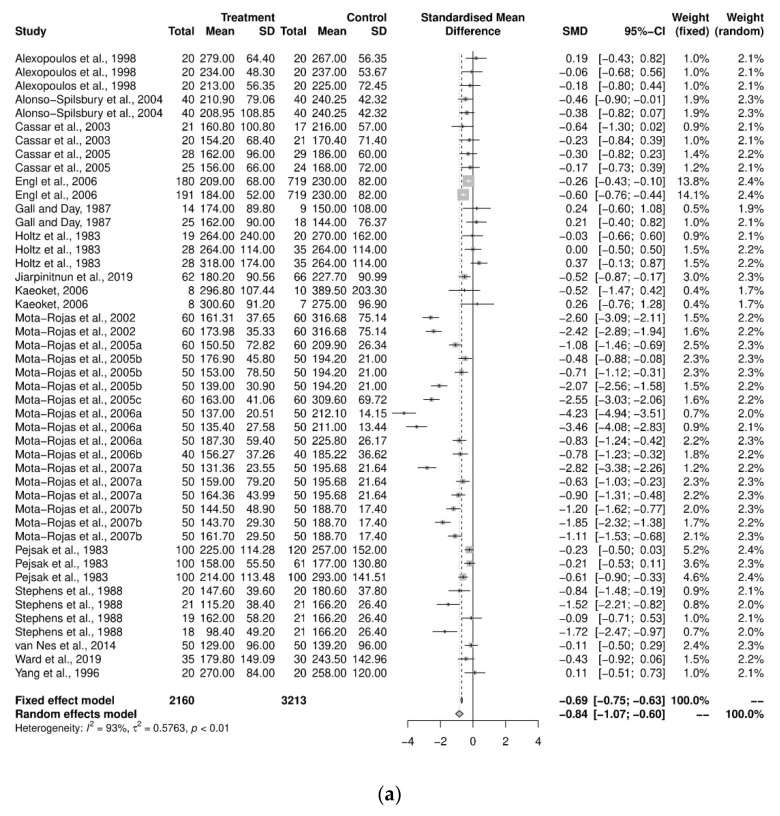
(**a**) Meta-analysis examining mean farrowing duration: pooled analysis [9,10,11,12,13,14,15,16,17,18,19,20,21,28,30,31,33]. (**b**) Meta-analysis examining mean farrowing duration: sub-group analysis by study objective (induction program and farrowing assistance [29,30,31,32,34,35,36,37,38,39,40,41,42,43,44,45,46,47,48,50,51,53].

**Figure 6 animals-12-01795-f006:**
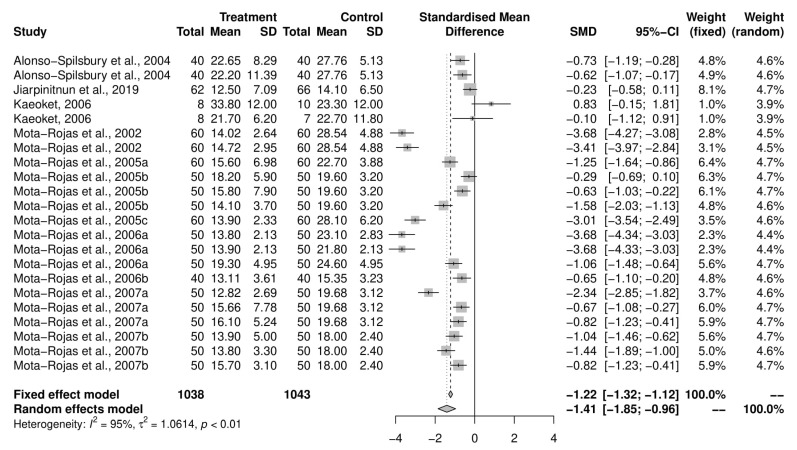
Meta-analysis examining mean birth interval between piglets [30,37,38,39,40,41,42,43,44,45,46].

**Table 1 animals-12-01795-t001:** Summary of studies included in meta-analyses of mean stillbirth, farrowing duration, and/or interval between piglets. Check mark (✓) indicates which studies were included in each meta-analysis.

Study	Study Objective	Stillbirth	Farrowing Duration	Interval between Piglets	Risk of Bias
[29]	Induction program	✓	✓		Some concern
[30]	Farrowing assistance	✓	✓	✓	Some concern
[31]	Induction program		✓		Some concern
[32]	Induction program		✓		Some concern
[33]	Induction program	✓			Some concern
[34]	Induction program		✓		High Risk
[35]	Induction program	✓	✓		Some concern
[36]	Induction program	✓	✓		Some concern
[37]	Farrowing assistance	✓	✓	✓	Some concern
[38]	Induction program	✓	✓	✓	Some concern
[39]	Farrowing assistance	✓	✓	✓	Some concern
[40]	Farrowing assistance	✓	✓	✓	Some concern
[41]	Farrowing assistance	✓	✓	✓	Some concern
[42]	Farrowing assistance	✓	✓	✓	Some concern
[43]	Farrowing assistance	✓	✓	✓	Some concern
[44]	Farrowing assistance	✓	✓	✓	Some concern
[45]	Farrowing assistance	✓	✓	✓	Some concern
[46]	Farrowing assistance		✓	✓	Some concern
[47]	Farrowing assistance		✓		Some concern
[48]	Induction program	✓	✓		Some concern
[49]	Induction program	✓			Some concern
[50]	Farrowing assistance		✓		Some concern
[51]	Farrowing assistance	✓	✓		Some concern
[52]	Induction program	✓			Some concern
[53]	Induction program	✓	✓		Some concern

## Data Availability

Not applicable.

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
