# Peer review of "Defining the Effect of Oxytocin Use in Farrowing Sows on Stillbirth Rate: A Systematic Review with a Meta-Analysis"

_animals, 2022, doi:10.3390/ani12141795_

Round 1

Reviewer 1 Report

The use of oxytocin during farrowing in pigs is a controversial topic due to its benefits and adverse effects on the sow and the fetus. This meta-analysis provides an extensive study of the available data regarding its use and the presentation of stillborn. I have left some comments hoping they can help the authors.

Line 49: ¿What type of stillborn did the authors consider? Please, specify.   

Line 31: Do you consider that 24 studies are enough to compare information for the meta-analyses?

Line 56: I suggest including other maternal or fetal factors that can influence the presentation of stillborn

Line 57: You could be more precise and indicate in which region of the hypothalamus it is produced.

Lines 91-92: These citations are not found in the reference list. Likewise, revise the citation style.

Line 134: Please, include the keywords used for the literature search.

Line 223: We suggest that you may also consider other characteristics like congestion, edema, or umbilical cord hemorrhage and not only rupture.

Line 236: I marked this specific line, but throughout the manuscript, several citations are written with superscripts. Please, amend accordingly. For example: lines 284, 295. In other sections, the citations are between parenthesis, in bold letters, etc.

Figure 1: I would suggest changing “N” to “n” inside the figure. Also, the picture is way too small to be able to read the methodology. But that will probably be changed after the editing process. Additionally, the sentence “The intervention is not the main focus of the s….” seems to be incomplete.

Line 336: Why do you think that in Mexico the use of oxytocin has been greater compared to other countries?

Line 339: Did the authors explore what type of commercial farms were evaluated in the studies included in the meta-analysis? If they were intensive or extensive systems? I consider this to be a relevant aspect that can be mentioned.

Line 352: I consider that Table one should be placed right after this paragraph. Also, please, revise the citation style inside tables. Most of the time, as in the text, it must be cited by numbers and not “authors + year”

Line 356: I think the authors wanted to write “bias” instead of “bis”

Line 426: Please, specify what the authors mean by “reduced number”.

Figure 2-6: I would suggest including only one or two decimals in the analyses.

Line 546: I consider it relevant to briefly discuss the endocrine basis of why induction programs use prostaglandin before oxytocin administration. If every study or every farm uses both, could prostaglandin also influence the farrowing process?

Line 551: Please, insert a reference.

Line 556: The number 5 is found as a superscript. Please correct the format. Likewise, assess whether the information in brackets is relevant.

Lines 559-562: I would suggest delving into the physiological explanation of these results. For example, the effect of oxytocin on the myometrium.

Lines 578-585: I think it would be interesting to discuss the importance of the time intervals between both drugs (prostaglandin and oxytocin). It may be something related to its pharmacokinetic characteristics such as plasma peak concentrations, or even the number of available oxytocin receptors at certain periods of the parturition stages.

Lines 610-614: I would suggest mentioning the importance and possible application of this knowledge, emphasizing under what circumstances a benefit could be obtained with the administration of oxytocin.

Lines 629-632: You wrote: “The meta-analyses demonstrated that sows that received oxytocin had an increased number of stillborn piglets per litter but reduced farrowing duration and birth interval between piglets compared to control sows”. Could you please explain according to reviews why this is happening?

References: Amend style according to the instructions for author’s guide.

Decision: Accepted with major changes.

-           

Reviewer 2 Report

Line 324: remove "written by a person known to submit false research papers to journals"  Remove the citation as well

The authors did a nice job at the end of the discussion section addressing the use of oxytocin for dystocia as this was a question of the reader throughout the manuscript.  Timing of the use of oxytocin after the initiation of parturition is a factor of interest and could be expanded upon beyond the discussion.

Author Response

Line 324: remove "written by a person known to submit false research papers to journals"  Remove the citation as well

The citation has been removed and the text altered accordingly.

The authors did a nice job at the end of the discussion section addressing the use of oxytocin for dystocia as this was a question of the reader throughout the manuscript.  Timing of the use of oxytocin after the initiation of parturition is a factor of interest and could be expanded upon beyond the discussion.

We have added:

The practical implications of this review is that oxytocin given early in the farrowing process to sows regardless of whether or not they are having difficulties, results in shorter farrowing duration but increases the likelihood of a stillborn piglet, whereas oxytocin given later in the farrowing process to sows with dystocia likely results in fewer stillborn piglets compared to no intervention.

Reviewer 3 Report

Very comprehensive study on the use of oxytocin in farrowing sows.

The figures should be improved, maybe you can change the way it is presented to make the figures more visible. 

line 356 - Among the 46 references, risk of bias assessments

line 371 - in head of Table 1 should be "Interval between piglets" written

1.      What is the main question addressed by the research?

The main research question is whether there are any undesirable side effects when using oxytocin in farrowing sows.

2. Do you consider the topic original or relevant in the field, and if so, why?

The topic is definitely relevant because the administration of oxytocin has been used as a routine procedure in farrowing sows for many years without fear of adverse effects. Moreover, the use of hormones today is also problematic from the point of view of animal welfare.

3. What does it add to the subject area compared with other published material?

The added value of this topic lies in the comprehensive analysis of many different sources used for it.

4. What specific improvements could the authors consider regarding the methodology?

I am not an expert on the methodology of meta-analyses, but the detailed description of the methodology and especially the risk of bias seems very informative.

5. Are the conclusions consistent with the evidence and arguments presented and do they address the main question posed?

Yes, definitely. The evidence and arguments could be accepted and support the main hypothesis about the existence of some adverse effects of the use of oxytocin.

6. Are the references appropriate?

In general, the references are adequate, although some of them belong to an interdisciplinary field (medicine, pharmacology,...).

Author Response

The figures should be improved, maybe you can change the way it is presented to make the figures more visible. 

We have tried to enlarge the figures and make them more visible. We are hoping that during the publishing process, we will receive assistances from the journal to have the format of the figures fixed.

line 356 - Among the 46 references, risk of bias assessments

The typo has been fixed.

line 371 - in head of Table 1 should be "Interval between piglets" written

Corrected.

Round 2

Reviewer 1 Report

The authors have answered every one of my observations. I am satisfied with your work. I have no more comments.

The article must be published.